



# Capturing functional strategies and compositional dynamics in vegetation demographic models

Polly Buotte[1], Charles Koven[2], Chonggang Xu[3], Jacquelyn Shuman[4], Michael Goulden[5], Samuel Levis[6], Jessica Katz[1], Junyan

Ding[2], Wu Ma[3], Zachary Robbins[7], Lara Kueppers[1]

[1]Energy and Resources Group, University of California Berkeley, Berkeley, CA, United States

[2]Climate and Ecosystem Sciences Division, Lawrence-Berkeley National Laboratory, Berkeley, CA, United States

[3]Earth and Environmental Sciences Division, Los Alamos National Laboratory, Los Alamos, NM, United States

[4]Climate and Global Dynamics, Terrestrial Sciences Section, National Center for Atmospheric Research, Boulder, CO, United States

[5]Department of Earth System Science, University of California Irvine, Irvine, CA, United States

[6]SLevis Consulting, LLC, Oceanside, CA, United States

[7]Forestry and Environmental Resources, North Carolina State University, Raleigh, NC, United States

*Correspondence to*: Polly Buotte (polly.buotte@berkeley.edu)

**Abstract.** Plant community composition influences carbon, water and energy fluxes at regional to global scales. Composition is a dynamic property of ecosystems, arising from complex feedbacks among the environment, disturbance, and plant physiology. Vegetation demographic models (VDMs) allow investigation of the effects of changing climate and disturbance regimes on vegetation composition and fluxes. Such investigation requires that the models can accurately resolve these feedbacks to

simulate realistic composition. Vegetation in VDMs is composed of plant functional types (PFTs), which are specified according to plant traits. Defining PFTs is challenging due to large variability in trait observations within and between plant types and a lack of understanding of model sensitivity to these traits. Here we present an approach for developing PFT parameterizations that are connected to the underlying ecological processes determining forest composition in the mixed-conifer forest of the Sierra Nevada Mountains of California, USA. We constrain multiple relative trait values between PFTs, as opposed

to randomly sampling within the range of observations. An ensemble of PFT parameterizations are then filtered based on emergent forest properties meeting observation-based ecological criteria under alternate disturbance scenarios. A small ensemble of alternate PFT parameterizations is identified that produces plausible forest composition, and demonstrates variability in response to disturbance frequency and regional environmental variation. Retaining multiple PFT parameterizations allows us to quantify the uncertainty in forest responses due to variability in trait observations. Vegetation composition is a key emergent

outcome from VDMs and our methodology provides a foundation for robust PFT parameterization across ecosystems.



# 1 Intro

Plant community composition has important influences on carbon, water, and energy fluxes at regional (Wullschleger et al., 2014) and global scales (Bonan, 2008). Because climate-driven shifts in community composition have occurred over the past

century (Adams et al., 2010;Allen and Breshears, 1998;Millar and Stephenson, 2015;Kelly and Goulden, 2008) and are projected to continue into the future (Buotte et al., 2018;Williams et al., 2013;Thorne et al., 2017), capturing compositional changes is critical for simulating feedbacks within the Earth System. Vegetation demographic models (VDMs) can resolve the ecological mechanisms that determine community composition and are computationally efficient enough to be coupled to Earth System Models (Fisher et al., 2018). These models track plant size, height, and canopy position, which allows for light competition,

competitive exclusion, and dynamic recovery after disturbance (Fisher et al., 2018).

Community composition is a dynamic property of ecosystems, arising from complex interactions among climate, disturbance, plant ecological strategies, and abiotic conditions (Johnstone et al., 2016;Stephenson, 1990). VDMs predict composition by simulating the effects of the environment on growth and mortality rates based on physiological functions, which are simplified by grouping species into plant functional types (PFTs) (Fisher et al., 2018;Koven et al., 2020;Lebauer et al., 2013).

Increasing the ecological resolution represented by PFTs can improve simulated vegetation-climate feedbacks (Druel et al., 2017) and ecotone transitions (Baudena et al., 2015).

The functional complexity of VDMs comes with inherent challenges. While VDMs have flexibility in PFT definitions (Fisher et al., 2015;Medvigy et al., 2009), observed trait variability, even within a single species, can be large (Kattge et al., 2020) and model sensitivity to these traits is not well understood. In addition, PFT parameterizations are likely to exhibit

variability across climatic gradients. It remains a challenge, given the non-linear feedbacks among climate, disturbance, and PFT composition in VDMs, to define PFT parameterizations that lead to accurate resolution of the interactions that determine community composition.

Prior research suggests that model parameter sensitivity is likely to vary according to the primary constraints on plant growth (Nemani et al., 2003) and disturbance regimes. In the mesic temperate forest, temperature has a strong effect on the

distribution of evergreen and deciduous broadleaf trees (Xie et al., 2015), and the simulated biome boundary between cold-deciduous hardwood and evergreen needleleaf trees was sensitive to temperature effects on leaf lifespan (Fisher et al., 2015). Competition for light exerts a strong control on tropical forest community composition (Farrior et al., 2016;Condit et al., 2013) and simulated coexistence of tropical PFTs depended on parameters that influence the relative differences in canopy tree growth and mortality rates (Koven et al., 2020;Massoud et al., 2019;Powell et al., 2018). In semiarid temperate forests, light availability,

water availability, and the fire regime exert important controls on forest composition (North et al., 2016;Nemani et al., 2003). However, the controls on forest composition within VDMs have not been examined in this forest type.

Here we present an approach for defining PFT parameterizations that ensures simulated forest composition is a result of the interactions among the ecological strategies the PFTs represent, alternate disturbance regimes, and climate. We illustrate this approach by defining two conifer PFTs in the Functionally Assembled Terrestrial Ecosystem Simulator (FATES) for the mixed

conifer forest of the Sierra Nevada Mountains, California USA. We define a pine PFT, representative of a shade-intolerant, moderately drought-tolerant, fire-resistant conifer and an incense cedar PFT, representative of a shade-tolerant, very drought-tolerant, less fire-resistant conifer. In this ecosystem, FATES simulations with robust PFT parameterizations should demonstrate 1) PFT-specific trait parameters related to shade-tolerance, drought-tolerance, and fire-resistance influence forest composition via their controls on growth and mortality rates; 2) forest composition is sensitive to the simulated fire regime through fire's





70 effect on the light environment and direct mortality; and 3) forest composition is sensitive to variation in water availability, with less pine in areas with low water availability compared with greater water availability.

## 2 Methods

### 2.1 Modeling Framework

FATES was developed through integration of the Ecosystem Demography (ED) model (Medvigy et al., 2009;Moorcroft et al.,
75 2001) with the Community Land Model (Oleson et al., 2013), with initial testing focused in Eastern U.S. forests (Fisher et al., 2015) and Panama tropical forest (Koven et al., 2020;Massoud et al., 2019). FATES resolves vegetation demographics at the level of the cohort, which represents the density of individuals of a given PFT, size, and canopy position. PFTs are defined by functional traits that describe plant physiology (e.g., photosynthesis, respiration, carbon allocation and turnover) and sensitivity to disturbance and environmental variation. Patches can contain multiple cohorts of plants and patch age is tracked according to
80 time since last disturbance. The number of patches and cohorts is dynamic during a simulation. Allocation of carbon to reproduction creates new cohorts within a patch. Disturbances caused by tree mortality, fire or harvest splits existing patches to create a new patch. Growth rates for each cohort are determined by carbon assimilation and allocation, which are affected by light and water availability and climate. Mortality is based on fire, carbon starvation, hydraulic failure, and cold-stress, along with a background mortality rate representing mortality sources not yet incorporated into the model. FATES computes
85 physiological processes on half-hourly time-steps, and growth, mortality, regeneration, and disturbance on daily time-steps. Here we have coupled FATES to the Community Land Model version 5 (Lawrence et al., 2019), which allows for a dynamic relationship between soil water availability and evapotranspiration that is governed by PFT water stress tolerance and soil physical properties. A full description of physiological and demographic processes in FATES can be found in Fisher et al. (2015), Koven et al. (2020) and the FATES Technical Note online at 10.5281/zenodo.3517271.

90  The simulation of wildfire in FATES is adapted from SPITFIRE, a forest fire behavior and effects model meant for use at regional to global scales (Thonicke et al. 2010). As implemented in FATES, fires are initiated based on a lightning ignitions dataset (Li et al., 2013) and once ignited are modulated for climate control with the Nesterov fire danger index. Fire behaviors, including rate of spread, duration, and intensity, depend on six classes of ground fuels and their moisture status. Scorch height is estimated for each cohort of trees, determining crown damage. Cambial damage, which is modulated by traits such as bark
95 thickness, canopy damage and cambial heating determine the probability of tree mortality. The amount of biomass consumed is calculated based on fire intensity and rate of spread.

### 2.2 Study Area and Forest Type

We simulated the two dominant conifer genera in California's mixed conifer forest: pine and incense cedar. The pine species in this forest, including ponderosa (*Pinus ponderosa*), Jeffrey (*Pinus jeffreyii*) and sugar (*Pinus lambertiana*) pine, are shade
100 intolerant and highly resistant to fire (North et al., 2016). Incense cedar (*Calocedrus decurrens*) is more shade- and drought-tolerant, but less fire-resistant (North et al., 2016). Surface fires, and the creation of microclimates suitable for pine regeneration are thought to be important for promoting pine dominance in the Sierra Nevada (Van de Water and Safford, 2011;Yeaton, 1983).

  We conducted a parameter sensitivity analysis and developed PFT parameterizations with FATES simulations at the Soaproot Saddle flux tower site (O'Geen et al., 2018). We evaluated simulated forest composition, model biases, and





environmental controls on coexistence across a regional domain that is dominated by the combination of pine (ponderosa, Jeffrey, and sugar) and incense cedar according to data produced by the Landscape Ecology, Modeling and Mapping Analysis (LEMMA) project (Ohmann et al., 2011) (Figure 1).



Sample observations maintaining trait correlations

Single site

*Ensemble of hypothetical PFT parameterizations*

Parameter sensitivity analysis

Fig. S1

Single site

*Ensemble of potential pine and incense cedar parameterizations*

Filter by ecological expectations (Table 1)

Fig. 2

Sample observations with trait correlations AND enforce between-PFT trait constraints

Fig. 3

Single site

*Ensemble of potential pine and incense cedar parameterizations*

Filter by ecological expectations (Table 1)

Quantify uncertainty due to observed trait variability

Fig. 4

*Ensemble of retained pine and incense cedar parameterizations*

Model Evaluation

Explore model biases

Environmental sensitivity analysis

Quantify uncertainty due to observed trait variability

Fig. 5
Fig. S2

Fig. 6

Fig. 7-10
Fig. S3-S6

Text



**Figure 1. Overview of our workflow for developing and applying PFT parameterizations. We suggest sampling with relative trait value**
**constraints to create the initial ensemble of potential PFT parameterizations. An initial sensitivity analysis may be necessary if**
**applying the VDM in a new ecosystem. Ecological expectations are developed according to understanding of the climate and**
**disturbance controls on coexistence in that ecosystem. These expectations are used to filter the simulation outcome, thereby retaining**
**PFT parameterizations that conform to their intended ecological niches. The retained parameterizations are applied to a regional**
**domain to evaluate model performance, model biases, and environmental controls, which can indicate potential for improvements to**
**PFT definitions or forcing data, or representation of processes within the model. Retaining an ensemble of parameterizations allows**
**for quantification of uncertainty in simulated outcomes due to variability in trait observations.**

## 2.3 Trait Data

We compiled a database of trait observations by tree species, starting with the TRY database (Kattge et al., 2011) and supplementing with data from additional literature where necessary (included in data archive https://doi.org/10.6078/D15M5X).
To limit variability in trait values resulting from diverse geographic locations, we focused our literature search on California, and 72% of the collected pine and cedar trait observations came from studies conducted in the Sierra Nevada Mountains. The remaining observations were from elsewhere in the Western US. We queried existing databases for allometric observations (Jenkins et al., 2004;Chojnacky et al., 2014;Falster et al., 2015).

## 2.4 Experimental Design and Analysis

Our approach combines of observations of plant traits and changes in forest composition under different disturbance scenarios with ensembles of model simulations to select robust parameterizations (Figure 1). After an initial parameter sensitivity analysis, we filter an ensemble of potential PFT parameterizations based on ecological criteria at a single site. We then evaluate simulated forest composition in the ensemble of retained parameterizations across a regional domain and explore model biases and environmental controls on composition and PFT-specific vital rates to suggest avenues for improving simulated forest
composition.

All FATES simulations were forced with 4x4 km spatial resolution daily climate data from 1979-2009 (Abatzoglou, 2013) disaggregated to 3-hourly intervals (Rupp and Buotte, 2020). Soil texture and organic carbon content were taken from the best available soils data for our domain, as described in Buotte et al. (2018), and, due to a lack of adequate spatially resolved soil data and no representation of root access to regolith water sources in FATES, soil depth was set to 10 m for all grid cells (O'Geen
et al., 2018;Klos et al., 2018).

Because FATES had not been previously exercised in the temperate mixed conifer forest, we assessed the sensitivity of simulated coexistence to 46 PFT trait and model parameters (Table S1). We defined two hypothetical PFTs, with trait values (Table S1) drawn from distributions of trait observations of all conifer species present at the flux tower site (SI trait database), to create a 720-member ensemble of FATES parameterizations. We randomly sampled the parameter space based on Latin
Hypercube sampling. We first divided each parameter range into intervals with equal probability and randomly sampled values from these intervals. We then ordered the sampled parameter values to maintain specified rank correlation among different parameters (Xu and Gertner, 2008). Some parameters, such as the target carbon allocated to storage reserves, are not observable; others are observable but regionally specific data are scarce or non-existent. For such parameters, ranges were determined based on previous sensitivity studies (Fisher et al., 2015;Koven et al., 2020;Massoud et al., 2019). We started these simulations from
bare ground and ran the ensemble for 100 years with fire active, recycling the 1979-2009 climate forcing.





We quantified composition as the ratio of the basal area of PFT #1 to the total basal area. This ratio therefore varies between 0, indicating complete PFT #2 dominance, to 1, indicating complete PFT #1 dominance. We used univariate, non-linear generalized additive models to quantify the variance in composition explained by the differences between PFT #1 and PFT #2 parameter values. Because each parameter is varied over its full range of realistic values, variable importance as measured by $R^2$
(coefficient of determination, or variance explained) is also a measure of parameter sensitivity.

Next, we created an ensemble of parameterizations for a shade-intolerant, fire-resistant pine and a shade-tolerant, drought-tolerant, less fire-resistant incense cedar (Table S1). The parameter sensitivity results, along with the availability of observations, informed our decision of which trait parameters to vary. We varied eight trait parameters to capture the differences in these two ecological strategies. We represented plant response to the light environment with four trait parameters: the specific
leaf area at the top of the canopy (SLA top), the maximum possible specific leaf area (SLA max), the maximum rate of carboxylation (Vc max), and leaf nitrogen (leaf N), which affects leaf respiration in FATES. The soil matric potential at which stomata close (SMPSC) controlled drought tolerance, and bark thickness (bark) controlled fire resistance. We varied two additional trait parameters, leaf lifespan (leaf life) and wood density (wood den), that differ between these two strategies (Niinemets, 2010;Kozlowski and Pallardy, 1997) but are not easily tied to light availability, water availability, or fire resistance
in FATES.

We constrained the eight trait parameter values to the distributions of observations of pine and incense cedar (Table S1), as opposed to the full range of conifer trait values as in the 720-member ensemble used in the parameter sensitivity analysis. All other trait parameters were held constant between the two PFTs as the mean of the combined pine and cedar observations. Although some of these trait parameters were found to be influential (e.g. allometric parameters), observations were insufficient
to distinguish between pine and incense cedar. Non-trait model parameters were set based on previous research with FATES (Table S1). Following the same sampling methods that maintain rank correlation between trait parameters as above, we created a 360-member ensemble of PFT parameterizations. We ran this ensemble for 100 years for a total of four scenarios: from bare ground and from initialized stands, with fire both active and inactive. Initialized stands began with an even proportion of pine and cedar, with the size structure based on census data from the flux tower site (included in data archive
https://doi.org/10.6078/D15M5X).

Observations allow us to devise ecological criteria, or expectations, for how the composition of trees with these two ecological strategies should respond to disturbance. To ensure the PFT definitions represented the intended ecological strategies, we filtered the ensemble of parameterizations based on eight criteria. In the mixed conifer forest of the Sierra Nevada, pine dominates when fire is present on the landscape (North et al., 2016), and incense cedar increases in dominance when fire is
excluded (Dolanc et al., 2014a;Dolanc et al., 2014b). From these observations we created six criteria based on pine and incense cedar basal area according to initial conditions and the presence of fire (Table 1). We included two criteria based on observations of leaf area index and carbon use efficiency (Table 1). We filtered the 360-member ensemble and retained ensemble members that met all eight criteria.

**Table 1. Ecologically expected outcomes for each disturbance (fire on vs off) and initial condition (bare ground vs initialized stands) scenario for FATES simulations at the Soaproot Saddle site in the Sierra Nevada Mountains of California. BA = basal area, NPP = net primary productivity.**

| FATES Scenario | Expected Conditions | Time Period/ Simulation Duration | References |
|---|---|---|---|
| Bare Ground start, Fire On | Pine BA > cedar BA Pine NPP > 0 | After 100 years | Dolanc et al. 2014 Soaproot Saddle census data |





| | Cedar NPP > 0 | | |
|---|---|---|---|
| Bare Ground start, Fire Off | Cedar BA > Cedar BA from bare ground with fire on<br>Pine BA < Pine BA from bare ground with fire on<br>Pine NPP > 0<br>Cedar NPP > 0 | After 100 years | Dolanc et al. 2014<br>Soaproot Saddle census data |
| Initialized start, Fire On | Pine BA > cedar BA<br>Pine NPP > 0<br>Cedar NPP > 0 | After 100 years | Dolanc et al. 2014<br>Soaproot Saddle census data |
| Initialized start, Fire Off | Cedar BA > Cedar BA from initialized with fire on<br>Pine BA < Pine BA from initialized with fire on<br>Pine NPP > 0<br>Cedar NPP > 0 | After 100 years | Dolanc et al. 2014<br>Soaproot Saddle census data |
| | LAI within 2-3 | Average 2009-2011 | MODIS, personal observations |
| | 0.32<Carbon use efficiency<0.58 | After 100 years | DeLucia et al. 2007 |

Shade-tolerant trees tend to have lower maximum rate of carboxylation (Vc max), lower dark respiration rates, higher specific leaf area (SLA), and longer leaf lifespan than shade-intolerant trees (Kozlowski and Pallardy, 1997;Niinemets, 2010). However, filtering the 360-member ensemble retained only one parameterization that preserved these relative trait parameter values for pine and incense cedar.

We therefore created a 72-member ensemble of pine and incense cedar parameterizations using the same eight trait
parameters varied in the 360-member ensemble, but further constrained to enforce the appropriate relative differences between these functional types (Table S1, Figure 1). Parameter values were drawn from pine and incense cedar trait observation distributions that were centered on the filtered parameterization from the 360-member ensemble, spanned one standard deviation of the mean, maintained between-trait correlations, and retained the appropriate relative differences between pine and incense cedar traits. This ensemble was run for 100 years for each of the four scenarios of initial stand conditions and fire at the flux
tower site. The results were filtered based on the eight criteria in Table 1 to identify the pine and incense cedar parameterizations most consistent with the eight expected ecological outcomes. This filtering retained four plausible parameterizations.

To evaluate performance of these parameterizations across a wide range of environmental conditions, we ran the four plausible parameterizations across our regional domain from bare ground with fire on, for 100 years. We compared the
simulated ratio of pine basal area to total basal area (hereafter referred to as pine fraction) with the LEMMA data (Ohmann et al., 2011), and evaluated area burned with data from the Monitoring Trends in Burn Severity (MTBS) data (Eldenshenk et al., 2007). We classified each FATES grid cell as having a reasonable pine fraction if it was within one standard deviation of the mean of the 30m LEMMA grid cells encompassed by the FATES grid cell. We compared simulated and observed annual area burned over the domain with probability density functions and boxplots of each distribution.

We evaluated model biases as the binary correct/not correct response as a multivariate, non-linear function of average annual temperature, total annual precipitation, and simulated annual area burned. Climate variables were averaged over the range of climate forcing data, 1979-2009, for each 4-km grid cell. We evaluated the environmental controls on simulated forest composition across the mixed conifer forest type in the Sierra Nevada. We statistically modeled the pine fraction as a multivariate, non-linear function of annual precipitation, average annual temperature, and soil characteristics (percent sand, clay,





and organic carbon). All statistical analyses were performed using the mgcv package (Wood, 2011) in R version 3.6.2 (R Core
Team, 2019).

# 3 Results

## 3.1 Sensitivity of PFT Composition to Trait and Model Parameters

Coexistence between two hypothetical conifer PFTs was most influenced by trait parameters controlling gross primary
productivity and carbon allocation, as controlled in part by allometry (Figure S1). Allometric parameters, and wood density, set
the growth rates of stem diameter and thus tree height growth per unit of biomass gained. Non-trait model parameters
controlling the creation of new patches from tree-fall (Disturb Frac), and height sorting to determine canopy position (Comp
Excln) were among the least important (Figure S1). We used these sensitivity results to focus further analysis on the influential
trait parameters that distinguish pine and cedar strategies, and ensure we held sensitive but observationally unconstrained
parameters constant between the two PFTs.

## 3.2 Constraining Potential Pine and Incense Cedar PFT Parameterizations

Only one of the 360 ensemble members had the appropriate relative differences in pine and incense cedar trait values and met all
eight ecological criteria. This single ensemble member was used as the center point for generating the 72-member ensemble in
which the relative trait parameter values for pine and incense cedar were additionally constrained according to the ecological
strategies represented by each PFT (Figure 2). When between-PFT constraints were not enforced in sampling the observations,
many ensemble members (grey points in Figure 2) fell outside of the range of relative trait values that represent these two
ecological strategies.

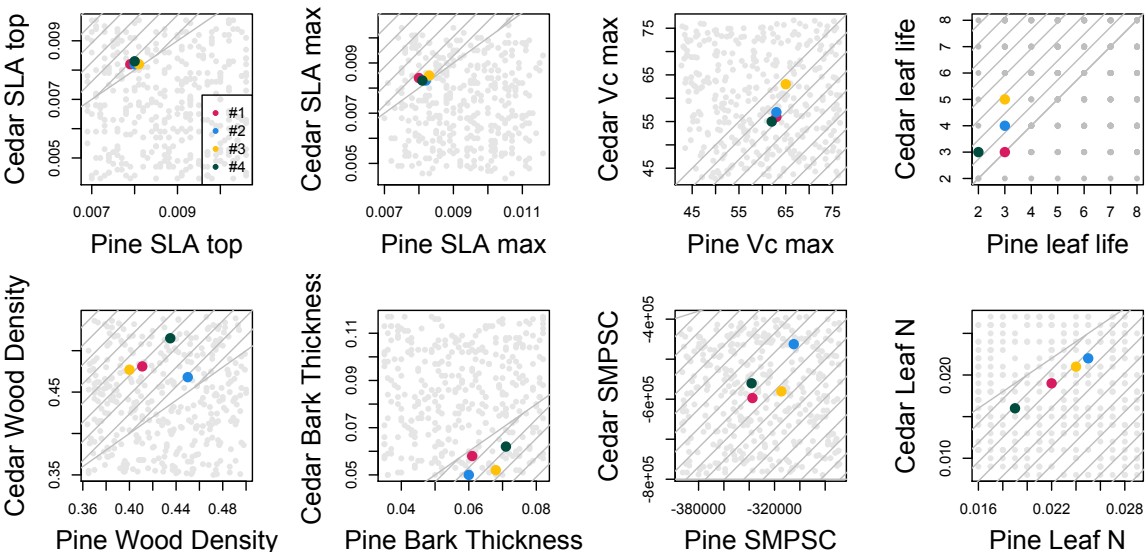

**Figure 2. Distribution of pine and incense cedar parameter values in a 360-member ensemble (light grey), in which selected values**
**were constrained by trait correlations and the distributions of observations, and in a 72-member ensemble (dark grey), in which values**



**were additionally constrained to preserve the appropriate relative values between pine and incense cedar (hatched area). The parameterizations retained from filtering based on expectations in Table 2 are shown in colors.**

Filtering the 72-member ensemble based on ecological criteria with and without fire was critical for selecting parameterizations that yielded the correct pine fraction under alternate fire regimes (Figure 3). While many ensemble members
(parameterizations) met individual ecological criteria, four members met all criteria regarding the effects of fire (Figure 3) and also were within the range of observed leaf area index and carbon use efficiency (not shown). After continuing simulations with these four parameterizations for another 100 years, all four still met the ecological criteria. Because these parameterizations span a range of observed pine and cedar trait values (Figure 2), they show differences in the magnitude of the effect of fire on the pine fraction (Figure 4). All four parameterizations show a decrease in pine fraction when fire is excluded (Figure 4), and all four
have the appropriate relative trait values (Figure 2). Retaining multiple, plausible PFT definitions allows us to quantify the uncertainty in simulated outcomes due to variability in trait observations. For example, when starting from even stands of pine and incense cedar, variability in observed traits leads to a 26-84% decline in the total pine fraction when fire is inactive (Figure 4). Taking canopy position into account, variability in observed traits leads to a 24-102% increase in the fraction of incense cedar in the canopy and 56-178% increase in the understory when fire is inactive (Figure 4).

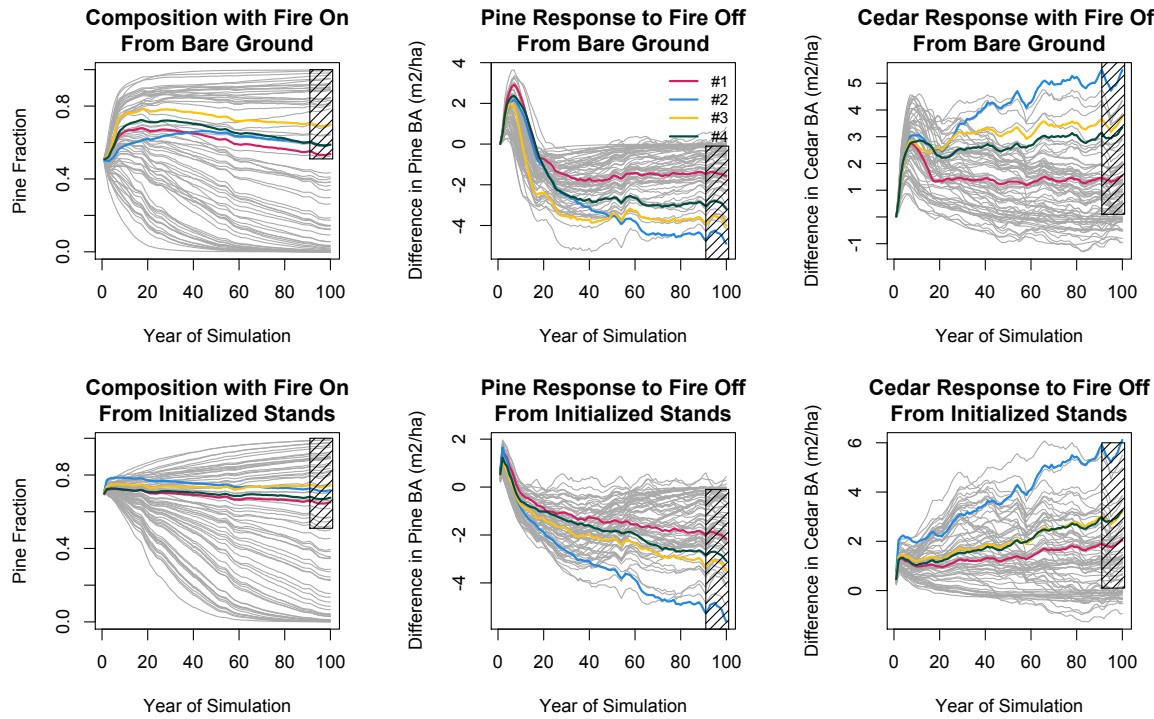


**Figure 3. Ensemble predictions relative to six of the filters based on ecological expectations listed in Table 2. Each simulation (lines) had a unique PFT parameterization. Black hatched areas indicate the range of expected outcomes. Green lines indicate simulations that were retained and grey lines indicate those excluded after applying all eight filters.**

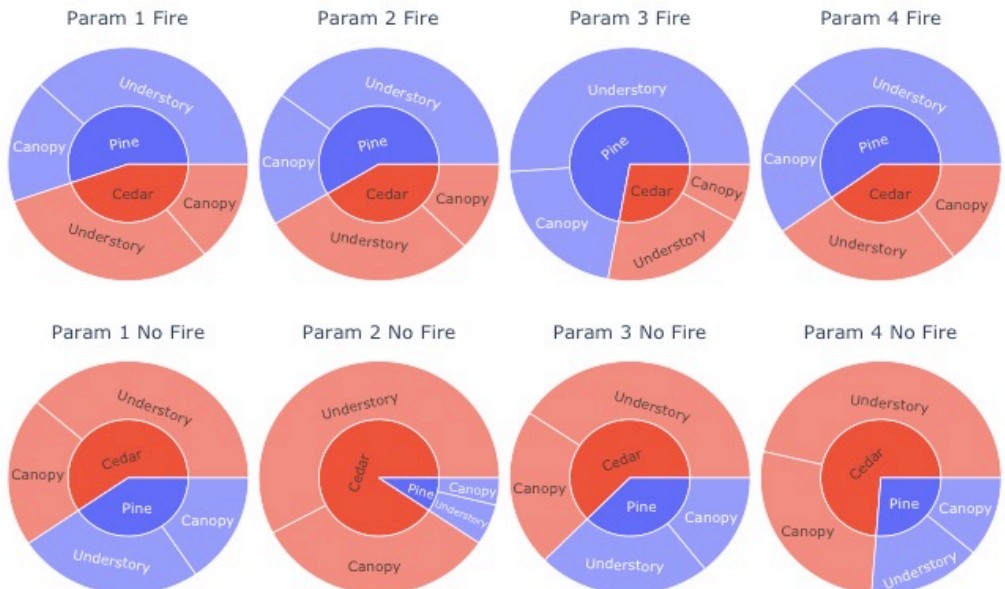

**Figure 4. Fraction of pine (blue) and incense cedar (red) basal area (inner circle), differentiated by canopy position (outer circle), in simulations at the Soaproot Saddle flux tower site. Each circle shows one of the four parameterizations retained after filtering based expectations in Table 1. Simulations were started from even stands and run with fire active (top row) and inactive (bottom row) for 100 years with recycled 1979-2015 climate.**

## 3.3 Evaluation of Regional Forest Composition

When we applied the 4-member ensemble of PFT parameterizations across the Sierra Nevada mixed-conifer domain, 79% of all grid cells were classified as having the correct (within one standard deviation of observed) ratio of pine to total basal area in all four simulations (Figure 5). In each simulation, over 85% of the incorrect grid cells under-represented pine basal area. Annual area burned and fire size were similar to observations, although FATES lacked representation of very large fires (Figure S2). Regression analyses indicated that all four parameterizations underestimated the pine fraction where precipitation was the lowest (Figure 6a) or area burned was the least (Figure 6b). The response functions for the other climate and environmental variables had 95% confidence intervals that spanned zero along the entire range of the independent variable, indicating they were not reliable predictors of FATES ability to simulate the correct pine fraction.

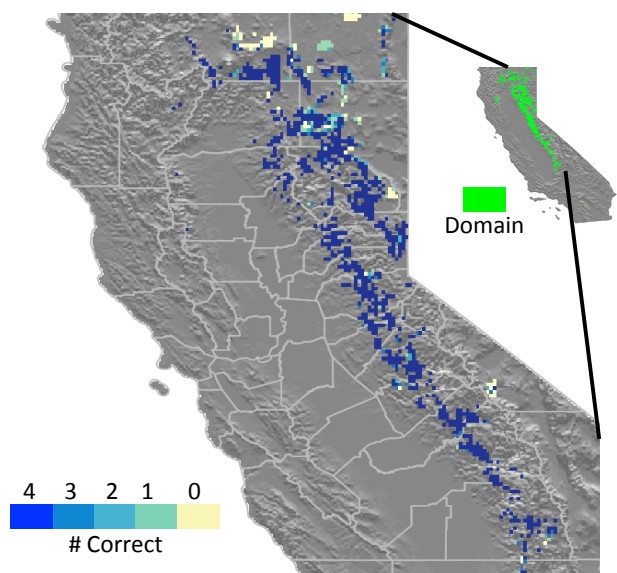

**Figure 5. Number of simulations with correct pine fraction, according to the LEMMA (Ohmann et al., 2011) dataset, out of four simulations with plausible pine and incense cedar parameterizations.**

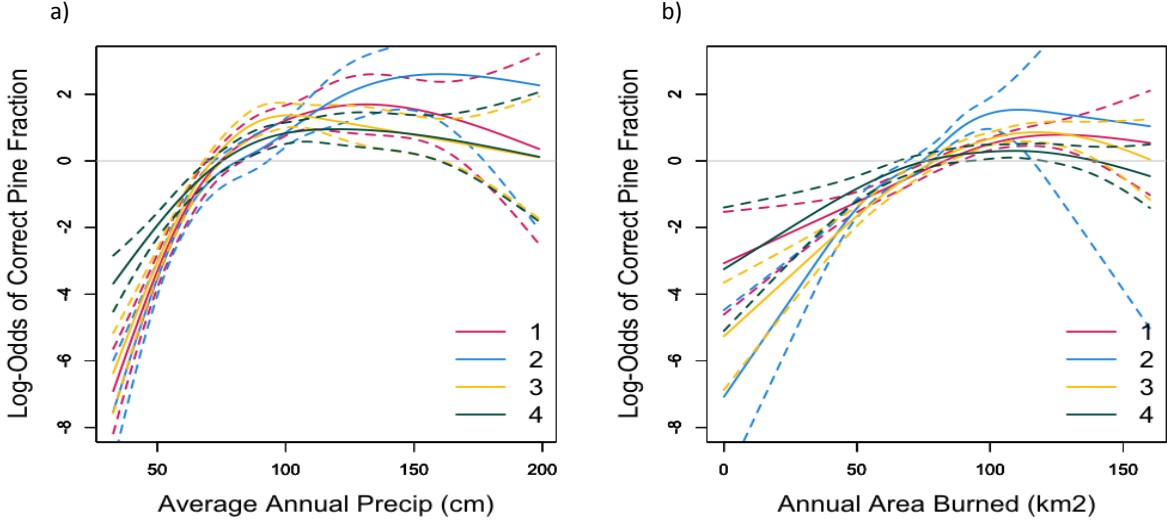

**Figure 6. Effects of average annual precipitation (a) and simulated average annual area burned (b) on the log-odds of the correct simulated pine fraction.**

## 3.4 Environmental Controls on Forest Composition

Regional variation in forest composition was most sensitive to precipitation (Figure 7a). Pine dominated in the wetter areas, with extreme incense cedar dominance in the driest areas in three of the four parameterizations (Figure 8a), this was not formally enforced by the eight expectations, but instead emerges from the combination of model dynamics with the eight enforced



expectations. Forest composition was less sensitive to soil characteristics, but cedar tended to dominate on soils with higher sand and clay content, and pine on soils with higher organic matter content (Figure 8b-d).


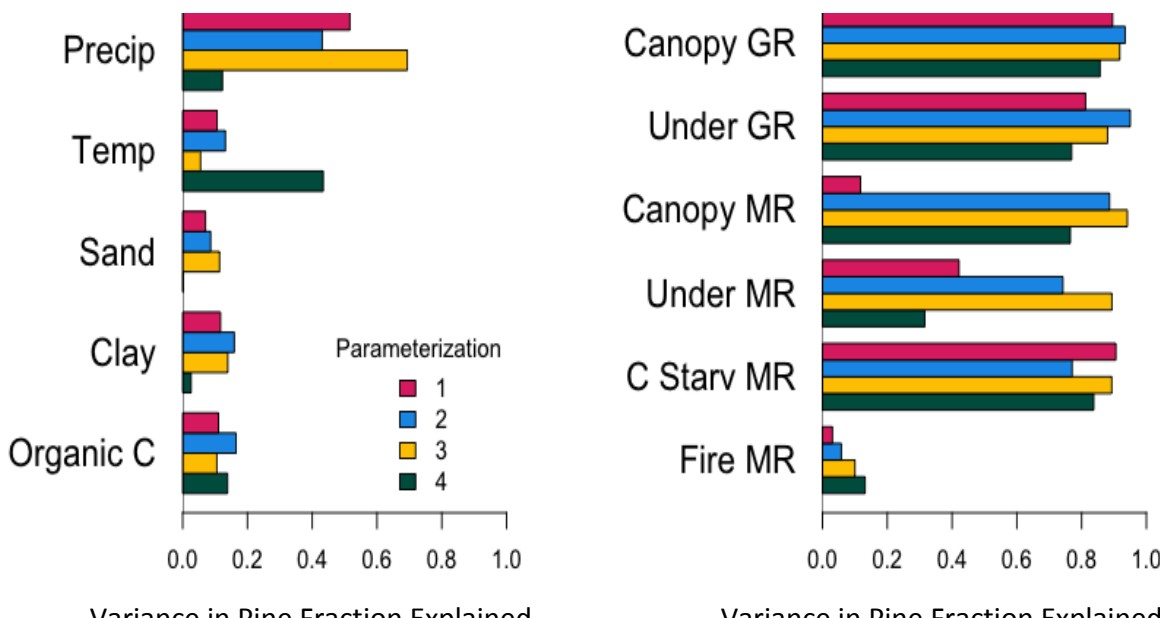

**Figure 7. Variance in the fraction of pine basal area relative to the total basal area of pine and incense cedar that is explained by environmental variables (a), and the difference between pine and incense cedar (pine minus cedar) growth and mortality rates (b), for each of four pine and cedar parameterizations over a regional domain in the Sierra Nevada mixed conifer forest, starting from bare**
**ground and run with fire active for 100 years. Pine fraction was calculated for the final year and rates were averaged over the duration of the simulations.**





**Figure 8. Effects of a) annual precipitation, b) soil sand content, c) soil clay content, and d) soil organic carbon content on the fraction of pine basal area relative to the total basal area of pine and incense cedar at the end of four simulations started from bare ground and run with fire active over a regional domain in the Sierra Nevada mixed conifer forest for 100 years. Each simulation uses one of the four parameterizations retained after filtering the outcomes of 72 parameterizations run at a single site according to the criteria in Table 1.**

Forest composition was sensitive to differences between pine and incense cedar vital rates (Figure 7b). PFT-differences in growth rates could be offset by opposite PFT-differences in mortality rates to prevent pine or incense cedar from excluding the other, and the degree of compensation possible varied among the four parameterizations (Figure 9). When pine growth rates were moderately faster than incense cedar, higher pine mortality rates allowed incense cedar to persist in the canopy and understory (Figure 9). Differences among trait values in the four parameterizations (Figure 2) allowed for varying degrees of compensation between growth and mortality rates (Figure 9). PFT-differences in growth rates were sensitive to precipitation and temperature (Figure S3). Pine growth rates were faster than incense cedar growth rates in wetter and cooler areas (Figure S4). Incense cedar growth rates were faster than pine growth rates in the driest areas, regardless of temperature (Figure S4).





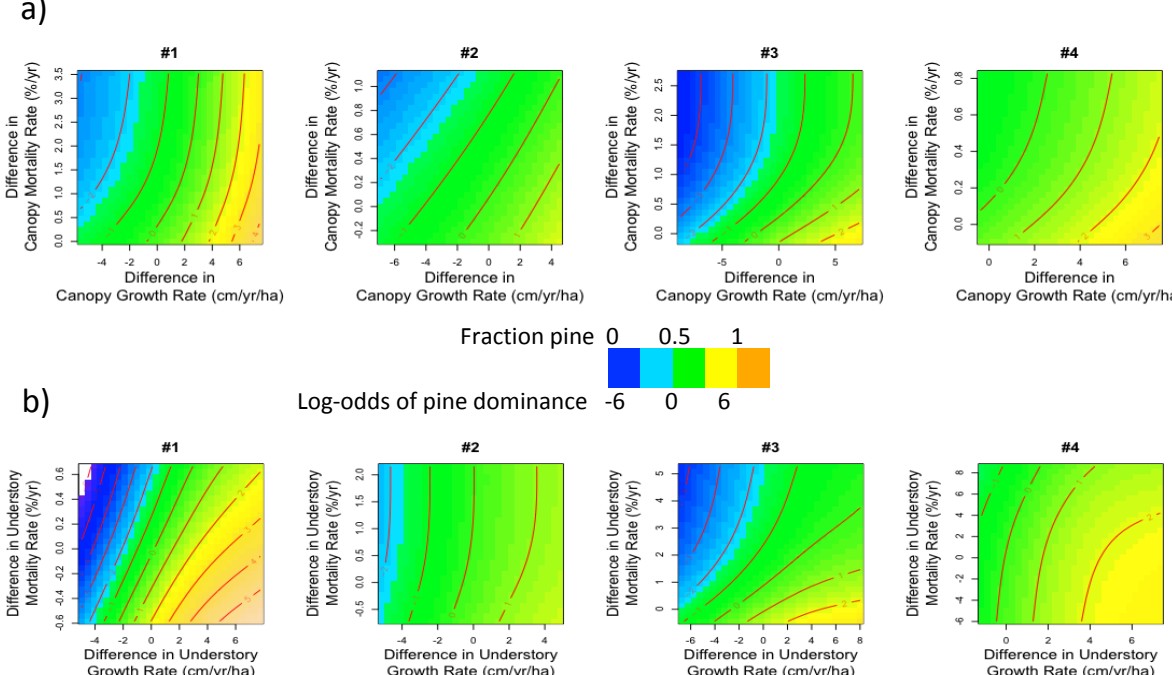

**Figure 9. Effects of differences (pine minus cedar) in (a) canopy and (b) understory growth and mortality rates on the fraction of pine basal area to the total basal area of pine and incense cedar at the end of four simulations started from bare ground and run with fire active over a regional domain in the Sierra Nevada mixed conifer forest for 100 years.**

Fire was the primary source of mortality across the mixed conifer forest domain in all four simulations (Figure S5). However, PFT-differences in fire mortality rates were less than PFT-differences in carbon starvation mortality rates (Figure 10), leading to PFT-differences in fire mortality rates having less influence on regional forest composition than PFT-differences in carbon starvation mortality rates (Figure 7b). Fire-caused mortality rates were similar between small pine and incense cedar trees, but higher for incense cedar among larger trees (Figure 10a). Pine had higher carbon starvation mortality rates across all

size classes (Figure 10b). PFT-differences in carbon starvation mortality rates were sensitive to climate (Figure S3), with a sharp increase in pine mortality at the lowest precipitation levels (Figure S6), where pine growth rates were much lower than incense cedar (Figure S4). PFT-differences in fire-caused mortality rates were less sensitive to climate (Figure S3).

a)

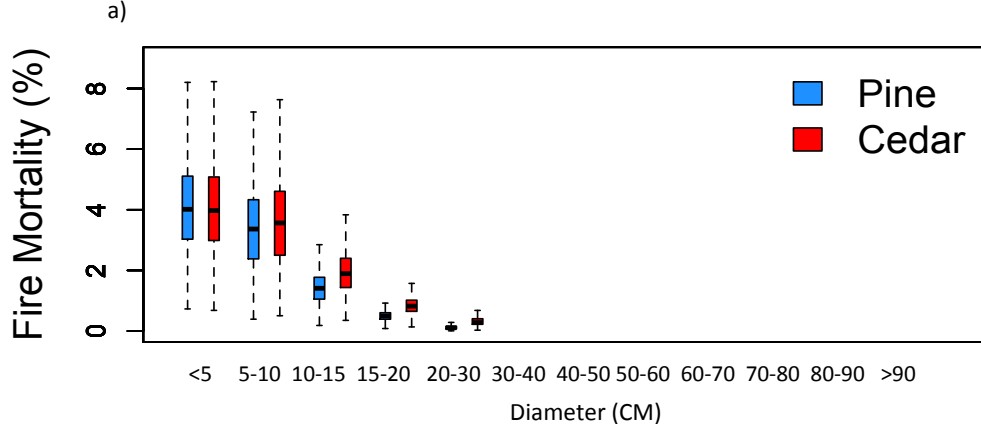

b)

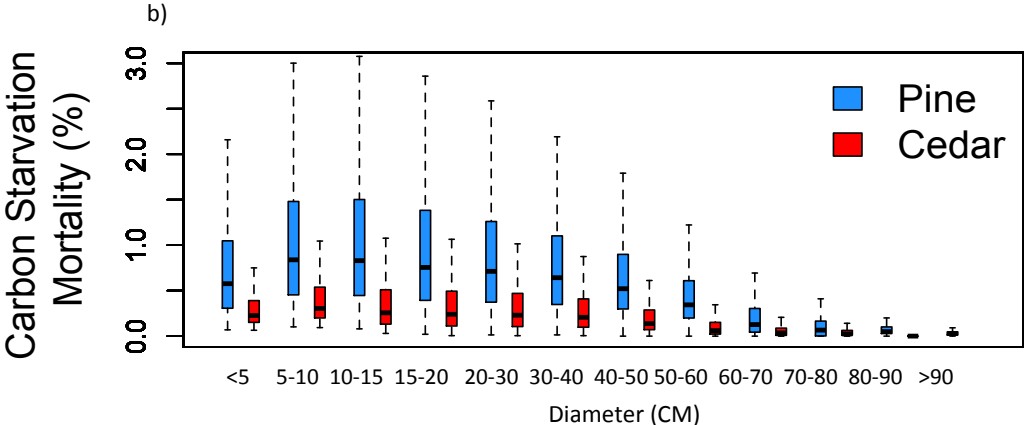

**Fig. 10. Pine and incense cedar fire mortality (a) and carbon starvation mortality (b) from smallest to largest size classes. Each box**
**plot shows the annual mortality over 100 years, pooled across the four simulations with the PFT parameterizations that met the**
**criteria in Table 2. Boxes bound the first and third quartiles, horizontal lines mark the median and whiskers extend to the max and**
**min values.**

# 4 Discussion

### 4.1 Approach for Defining PFTs

Creating PFT definitions that accurately resolve community composition is essential for simulating the Earth System (Wullschleger et al., 2014). We developed and applied a novel approach for assuring PFT definitions have high fidelity to the emergent properties of their intended ecological strategies. First we extended the common practice of sampling trait parameter observations based on observed correlations among traits within a PFT (Lebauer et al., 2013) by incorporating between-PFT parameter constraints. Secondly, we introduced an ensemble filtering process based on expected compositional changes in response to alternate initial condition and disturbance scenarios, and the emergent properties of leaf area index and carbon use efficiency. Finally, we evaluated the robustness of the resulting plausible PFT definitions across a wide range of environmental conditions, comparing simulations to observationally-constrained forest composition (Ohmann et al., 2011).



Several methods for parameter estimation are commonly employed, including Bayesian (Lebauer et al., 2013), maximum likelihood (Medvigy et al., 2009), and iteration (Hudiburg et al., 2009). However, these methods do not ensure that simulated composition, even when accurate, is a result of the mechanisms that drive composition (Williams et al., 2009). Employing between-PFT parameter constraints and filtering simulations by process outcomes connects the PFT definitions to the processes that drive community composition. In the mixed conifer forest of the Sierra Nevada, pine dominates when fire is present on the landscape (North et al., 2016), and incense cedar increases in dominance when fire is excluded (Dolanc et al., 2014a;Dolanc et al., 2014b). This knowledge allows us to create eight criteria based on pine and incense cedar basal area according to initial conditions and the presence of fire (Table 1). We found many PFT parameterizations that met one of these criteria. However, fewer parameterizations met all eight criteria. Further filtering based on ecophysiological constraints (here, carbon use efficiency) and emergent properties (here, leaf area index) provide additional connections to field-based understanding of how the ecological strategies we are representing interact to determine community composition.

Our understanding of the importance of constraining between-PFT parameter values emerged during the course of our analysis. Even though the trait parameter values in the 360-member ensemble were drawn from observations subject to observed within-PFT trait correlations, filtering retained only one parameterization with the appropriate relative pine and incense cedar values across all eight trait parameters. In contrast, filtering the 72-member ensemble, in which between-PFT constraints were applied, resulted in four plausible parameterizations and allowed us to quantify uncertainty in simulated forest composition due to variability in trait observations. A greater proportion of the potential parameters were retained in the 72-member ensemble because the between-PFT trait constraints ensured pine would respond to the environment as the less shade-tolerant, less drought-resistant, and more fire-resistant PFT as compared to incense cedar. Including between-PFT trait constraints ensures that the PFT responses to environmental conditions are in accordance with the ecological strategies the PFTs represent. The process would be more efficient if between-PFT constraints were enforced before filtering an ensemble, as depicted by the center box with heavy outline in Figure 1.

We developed a set of plausible PFT parameterizations at a single site, and then applied those parameterizations across a regional domain with variability in climate. Starting with FATES simulations at a single site allowed us to reduce the computational cost of simulating hundreds of potential parameterizations. Evaluating the retained parameterizations across the regional domain allowed us to use model biases to determine if the retained parameterizations were robust across temperature and precipitation gradients, and devise options for improving model performance.

Our approach could be easily applied in other ecosystems, with ecological expectations and scenarios developed in accordance with the accumulated knowledge of the controls on community composition. We suggest conducting an initial parameter sensitivity analysis to ensure influential parameters can either be estimated based on observations or held constant. In our 720-member ensemble, trait parameters were bounded by observations of all conifer species present at the site, ensuring trait parameters spanned a broad range, and thus limiting the potential for missing influential parameters due to a lack of variability. However, a sensitivity analysis could be run on the ensemble created by sampling with inter-trait and inter-PFT constraints instead.

Coupling VDMs to Earth System Models is providing new opportunities for global change research (Fisher et al., 2018), and defining global PFTs is a critical component of this integration. Current vegetation distributions are the result of particular sequences of climate, disturbances, and dispersal events (Jackson et al., 2009). Therefore, a global model's ability to replicate current vegetation distributions is perhaps an unrealistic benchmark. Functional relationships among climate, disturbances, and vegetation distributions may provide a more meaningful benchmark for simulations. Our strategy of filtering ensembles of potential parameterizations at single sites and then evaluating model performance and biases across larger domains





would be an efficient means of arriving at robust global PFT definitions. First, an ensemble of potential PFT definitions would
be created, maintaining the appropriate inter-trait and inter-PFT correlations. Next, sites could be selected to represent
conditions with known coexistence and known competitive exclusion among two or more PFTs. It may be useful to stratify sites
based on the limitations of temperature, radiation, and water (Nemani et al., 2003), and to capture distinct disturbance regimes.
Ecological expectations would then be developed for each site-PFT combination to filter the ensemble of potential PFT
definitions. The filtered parameterizations can then be evaluated across a larger domain with gradients of climate and soils to
determine if additional modifications are necessary before investing in global simulations.

### 4.2 Sierra Nevada Forest Composition


Enforcing the relative parameter constraints and filtering based on ecological criteria resulted in PFT definitions that led to
realistic emergent dynamics and forest composition that met all three of our driving expectations. Given the historical
occurrence of seasonal drought and frequent surface fires in the mixed conifer forest region of the Sierra Nevada Mountains
(North et al., 2016), we expected that composition of tree functional types in FATES would be sensitive to parameters related to
shade-tolerance, drought-tolerance, and fire-resistance. Our results described a simulated ecosystem where forest composition is
driven by available light and water and the presence of fire. Forest composition in FATES was sensitive to differences between
the PFTs in specific leaf area, $V_{c,max}$, and leaf respiration, reflecting the importance of the light environment (Kozlowski and
Pallardy, 1997). We found composition was also sensitive to variation in bark thickness. Within FATES (following Thonicke et
al 2010), thicker bark provides insulation against cambial damage from fire and thereby lowers tree mortality due to fire. Unlike
in tropical forest, composition was not sensitive to parameters that control patch creation from small-scale disturbances (Koven
et al., 2020), indicating the landscape-scale disturbance from fire was more important than disturbances such as tree fall. FATES
was not sensitive to differences in the parameter controlling the soil matric potential at which stomata close (SMPSC). However,
differences in PFT dominance according to precipitation and soil characteristics that define the water holding capacity indicate
water availability affected composition.

385           Simulated between-PFT differences in regional growth and mortality met our expectations of the influence of the fire
regime and water availability on forest composition, and increased our confidence in FATES' ability to represent the ecological
dynamics in the Sierra Nevada mixed conifer forest. Our filtering process forced the expected changes in pine and cedar
abundance due to fire. The emergent responses in growth and mortality, however, were not enforced yet conformed to our
expectations. When fire is active in the model, tree mortality from fire should open canopy gaps, increasing light availability and
favoring pine (Yeaton, 1983;North et al., 2016). Conversely, when fire is inactive, the canopy should close, reducing light
availability and favoring incense cedar (North et al., 2016;Dolanc et al., 2014a). The combination of increasing pine dominance
with increasing area burned, and increasing pine dominance with greater differences between pine and cedar growth rates
supports these expectations. Fire was the dominant source of mortality, with large incense cedars experiencing relatively greater
mortality from fire than pines did. Our filtering process did not force the expected pine and cedar dominance along the
precipitation gradient. Yet, our regional simulations reflect the expected drought-tolerance strategies: pine was more dominant
in wetter areas and pine growth rate was lower and carbon starvation mortality rate was higher than incense cedar in drier areas.

          Exploration of model biases across the regional ensemble, along with analyses of the environmental controls on forest
composition and between-PFT differences in vital rates revealed a deficiency in our current simulations in regards to water
availability. In all four parameterizations, pine was underrepresented at the lowest precipitation levels. This could indicate that,
given the range of observed variability in pine carbon allocation and drought tolerance (DeLucia et al., 2000), further delineation
of a dry pine PFT may be necessary to simulate this forest type across its full range in the Sierra Nevada. Another possibility is





that variability in root-depth distributions, in conjunction with improved soil definitions, may be necessary. Root distributions were held constant between the pine and cedar PFTs due to a lack of observations. Recent analysis with FATES at the Soaproot Saddle site (Ding et al., in revision) indicates that increasing rooting depth yields higher pine productivity in dry conditions

compared with shallow rooting depth. Alternatively, this model bias may indicate a structural deficiency in how drought stress is represented. In our simulations, water stress is represented with a scaling factor that reduces potential productivity (Oleson et al., 2013). Incorporating an explicit representation of the flow of water through the soil-plant-atmosphere continuum (Christoffersen et al., 2016) may be necessary to represent forest dynamics in a climate with strong seasonal drought. Further iterations of the process of defining PFTs and evaluating model biases with an additional PFT and variable rooting parameters

could indicate whether improved parameterizations or additional model processes are needed to correct this bias.

Our domain has historically experienced a surface fire regime (Van de Water and Safford, 2011;North et al., 2016). Our simulations represented a surface regime, with frequent, small fires in all parameterizations. However, canopy fuels are not included in the calculations of fire behavior and characteristics and observations indicate forest composition is changing in ways that may promote increases in canopy fire (Menning and Stephens, 2007). Given the important role of fire in filtering ensemble

members, fire behavior algorithms should be updated to allow for the inclusion of canopy fuels. As these changes may influence competitive ability, pine and incense cedar parameterizations may require further updates. Our approach provides an efficient, albeit computationally demanding, means of updating PFT definitions as new developments are incorporated into FATES.

## 5 Conclusions

Plant functional type definitions determine vegetation demographic models' ability to accurately simulate plant

composition. Traditional means of parameterization, such as iteration, do not guarantee ecologically robust PFT definitions, and can be extremely slow when many parameters interact to determine outcomes. Imposing between-PFT trait parameter constraints and filtering an ensemble of parameterizations based on a discrete set of criteria for outcomes under alternate disturbance or environmental scenarios ensures that PFTs are representing their intended ecological strategies. We applied this approach to define four plausible PFT parameterizations for a shade-intolerant, fire-resistant pine and a shade-intolerant,

drought-tolerant, less fire-resistant incense cedar. All four parameterizations produced robust simulations of forest composition across the mixed conifer forest in the Sierra Nevada Mountains. Analyses of parameter sensitivity and PFT-specific vital rates indicate FATES simulated the expected interactions among the fire regime and light and water availability in this ecosystem. This approach could be applied in any ecosystem, or scaled up to define global PFTs. Robust resolution of community composition will allow us to use VDMs to address important questions related to future climate and management effects on

forest structure, composition, and carbon storage and feedbacks within the Earth system.

## Data Availability

Data not otherwise referenced in the text is available at https://datadryad.org/stash/dataset/doi:10.6078/D15M5X. Source code for the Community Land Model version used here is available at https://zenodo.org/badge/latestdoi/344922587 and FATES code is available at https://zenodo.org/badge/latestdoi/344935673.





## Author Contribution

PB, LK, CK, and CX designed the research. JS and SL provided code modifications. MG provided plot census data. CX performed the trait sampling and PB performed the modelling and all subsequent analyses. PB prepared the manuscript with contributions from all authors.

## Competing Interests

The authors declare that they have no conflict of interest.

## Acknowledgements

Support for this research (or paper) is provided by the University of California Office of the President, Award LFT-18-542511: "The Future of California Drought, Fire, and Forest Dieback". This research used resources of the National Energy Research Scientific Computing Center, a DOE Office of Science User Facility supported by the Office of Science of the U.S. Department of Energy under Contract No. DE-AC02-05CH11231. We thank Tara Hudiburg for time on the Cheyenne supercomputer at the National Center for Atmospheric Research (Computational and Information Systems Laboratory, 2019) under NSF grant DEB-1553049; Ryan Knox, Rosie Fisher, Greg Lemieux, and Alex Hall for helpful discussions that provided ecological and technical insights; and Michael Xiao for creating Figure 4. CDK and JD acknowledge support by the Director, Office of Science, Office of Biological and Environmental Research, Regional and Global Model Analysis Program of the U. S. Department of Energy under Contract DE AC02 05CH11231 through the Early Career Research Program. J. Shuman was supported as part of the Next-Generation Ecosystem Experiments – Tropics, funded by the US Department of Energy, Office of Science, Office of Biological and Environmental Research.

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





**Figure Captions**

Figure 1. Overview of our workflow for developing and applying PFT parameterizations. We suggest sampling with relative trait value constraints to create the initial ensemble of potential PFT parameterizations. An initial sensitivity analysis may be necessary if applying the VDM in a new ecosystem. Ecological expectations are developed according to understanding of the 760 climate and disturbance controls on coexistence in that ecosystem. These expectations are used to filter the simulation outcome, thereby retaining PFT parameterizations that conform to their intended ecological niches. The retained parameterizations are applied to a regional domain to evaluate model performance, model biases, and environmental controls, which can indicate potential for improvements to PFT definitions or forcing data, or representation of processes within the model. Retaining an ensemble of parameterizations allows for quantification of uncertainty in simulated outcomes due to variability in trait 765 observations.

Figure 2. Distribution of pine and incense cedar parameter values in a 360-member ensemble (light grey), in which selected values were constrained by trait correlations and the distributions of observations, and in a 72-member ensemble (dark grey), in which values were additionally constrained to preserve the appropriate relative values between pine and incense cedar (hatched 770 area). The parameterizations retained from filtering based on expectations in Table 2 are shown in colors.

Figure 3. Ensemble predictions relative to six of the filters based on ecological expectations listed in Table 2. Each simulation (lines) had a unique PFT parameterization. Black hatched areas indicate the range of expected outcomes. Green lines indicate simulations that were retained and grey lines indicate those excluded after applying all eight filters.


Figure 4. Fraction of pine (blue) and incense cedar (red) basal area (inner circle), differentiated by canopy position (outer circle), in simulations at the Soaproot Saddle flux tower site. Each circle shows one of the four parameterizations retained after filtering based expectations in Table 1. Simulations were started from even stands and run with fire active (top row) and inactive (bottom row) for 100 years with recycled 1979-2015 climate.


Figure 5. Number of simulations with correct pine fraction, according to the LEMMA (Ohmann et al., 2011) dataset, out of four simulations with plausible pine and incense cedar parameterizations.

Figure 6. Effects of average annual precipitation (a) and simulated average annual area burned (b) on the log-odds of the correct 785 simulated pine fraction.

Figure 7. Variance in the fraction of pine basal area relative to the total basal area of pine and incense cedar that is explained by environmental variables (a), and the difference between pine and incense cedar (pine minus cedar) growth and mortality rates (b), for each of four pine and cedar parameterizations over a regional domain in the Sierra Nevada mixed conifer forest, starting 790 from bare ground and run with fire active for 100 years. Pine fraction was calculated for the final year and rates were averaged over the duration of the simulations.

Figure 8. Effects of a) annual precipitation, b) soil sand content, c) soil clay content, and d) soil organic carbon content on the fraction of pine basal area relative to the total basal area of pine and incense cedar at the end of four simulations started from





bare ground and run with fire active over a regional domain in the Sierra Nevada mixed conifer forest for 100 years. Each simulation uses one of the four parameterizations retained after filtering the outcomes of 72 parameterizations run at a single site according to the criteria in Table 1.

Figure 9. Effects of differences (pine minus cedar) in (a) canopy and (b) understory growth and mortality rates on the fraction of
pine basal area to the total basal area of pine and incense cedar at the end of four simulations started from bare ground and run with fire active over a regional domain in the Sierra Nevada mixed conifer forest for 100 years.

Fig. 10. Pine and incense cedar fire mortality (a) and carbon starvation mortality (b) from smallest to largest size classes. Each box plot shows the annual mortality over 100 years, pooled across the four simulations with the PFT parameterizations that met
the criteria in Table 2. Boxes bound the first and third quartiles, horizontal lines mark the median and whiskers extend to the max and min values.