# Peer review of "Capturing functional strategies and compositional dynamics in vegetation demographic models"

_Biogeosciences, 2021_

## Author Comment (AC1)

Response to reviewers
bg-2021-54

Reviewer #1
General comments:
This manuscript presents a novel approach for parameterizing demographic vegetation models, an extremely challenging problem in the ecological modeling literature and an increasingly important one as these models are increasingly used in global studies (and likely represent the future of Earth System models). Specifically, the manuscript parameterizes two conifer PFTs --- pine and cedar --- in the CLM-FATES model (including the SPITFIRE wildfire submodel) such that model behavior reproduces real ecological patterns in a mixed conifer forest in the Sierra Nevada Mountains, USA. The target ecological patterns are sensitivity of modeled forest composition to (1) parameters related to shade tolerance, drought tolerance, and fire resistance; (2) fire regime; and (3) water availability. The authors start from observed trait distributions based on TRY and other trait databases, and use these distributions generate a fairly large (720 member) ensemble for exploring sensitivity of predicted composition to parameter values. Then, the authors create more precise parameter distributions based on species-specific observations and trade-offs, fixing parameters to which modeled composition is relatively insensitive, and ran another ensemble (360 members) to generate a distribution of compositional outcomes over 100 years and covering four scenarios (bare ground vs. initial conditions; with vs. without fire). From this ensemble, the authors identified parameter combinations that matched expected ecological responses to fire regime and relative parameter rankings. Finally, the authors use the final realistic set of parameters to run regional simulations over the Sierra Nevada mountains, compare the resulting predictions of composition to observations, and evaluate the relationships between model biases and climate variables.
Overall, I found this to be an exceptional modeling study. The topic is very important --- parameter uncertainty is a major hurdle for VDMs, and this study shows how ecological process-based understanding can be effectively leveraged to reduce those uncertainties. The analysis is robust and thorough, and the presentation is top-notch, both in terms of visualizations and writing. The paper would have been compelling even if it had ended with the final ecologically-filtered parameterizations, and the further analyses of climate variables and mortality drivers from the regional simulations further sets this paper apart. The discussion is highly effective at placing the work in context. I only have a handful of minor suggestions (see detailed comments), none of which detract from my overall opinion that this paper can be published basically without any revision. Well done!

Detailed comments:
L49, "Model sensitivity to these traits is not well understood". --- There is some additional work on quantifying model sensitivity and uncertainty associated with parameters that may be worth citing here; e.g., Dietze et al. 2014 (DOI: 10.1002/2013JG002392); Raczka et al. 2018 (DOI: 10.1029/2018jg004504); Shiklomanov et al. 2020 (DOI: 10.1111/gcb.15164).

(Disclaimer: I am the lead author of the last study mentioned there. Normally, I try not to avoid pushing citations of my own work in reviews, but that paper directly justifies this study).

Thank you for the additional references. We found them helpful and have incorporated them into the Introduction and Discussion.

L53, "sensitivity is likely to vary according to primary constraints on plant growth" --- Yes, but this is also strongly dependent on model structure and assumptions; for instance, many vegetation models do not have a concept of nitrogen limitation, even though this is known to be an important constraint on growth in N-limited ecosystems. I might just add, "...constraints on plant growth as represented in the model" or something to that effect.

We agree and have added treatment of this idea here and in the Discussion.

L67--71, "In this ecosystem, FATES simulations with robust PFT parameterizations should demonstrate..." --- I really like this framing for a modeling study!
Figure 1 --- This is a really nice figure; very effective at conveying the overall workflow of the study.

Thank you.

Figure 2 --- Another very nice figure overall; very effective summary of the multidimensional parameter space. However, I had trouble distinguishing the light vs. dark gray values. I would recommend either using a different shape for the points, or labelling the panels (a, b, c, etc.) and identifying which ones had the 72 member ensemble.

The 72-member ensemble points were missing from this figure. They are now plotted and the contrast between light and dark grey is apparent.

Figure 3 --- Yet another very nice figure! I think it's fine as is, but one additional idea: Since many of the responses have similar and relatively well-behaved temporal trajectories, you could save some space (and potentially identify some interesting correlations) by just looking at the final outcomes (e.g., value at end of simulation, or average of the last 5 years). Eliminating the time dimension would allow you to plot some of these responses in multivariate space against each other, or against parameter values, both of which could reveal some interesting patterns.

We found it helpful to think about the trajectories of different ensemble members as we were digesting the results to check, for example, if the trajectories were diverging towards the end of the simulation and should therefore be continued, and to examine the rate a PFT gained dominance in each scenario. While we do not comment on these details, we prefer to show the full time series to allow other readers to make their own evaluations.

Section 4.1 --- Overall, I really enjoyed this discussion. However, it may be worthwhile to highlight the difference between calibrating models against PFT composition vs. plot/stand-level C pools and fluxes. The latter is the more common target for land surface models like CLM, probably because (1) aggregate stand-level

variables like net C flux are an easier target to hit thanks to the many pathways for doing so; (2) greater relevance to land-atmosphere interactions, the original driver behind land surface model development; and (3) the greater availability of data (e.g., remote sensing, flux towers) at those scales. The focus on composition in this study is relatively unique (especially for this class of model), which is a major selling point of the work. But, a challenge moving forward is trying to get both composition and biogeochemistry correct at the same time (c.f., Shiklomanov et al. 2020; DOI: 10.1111/gcb.15164). The authors might consider spending a few sentences on this topic here.

Thank you. We agree that benchmarking fluxes is important. We added a figure and text describing comparison of simulated and observed GPP and ET at the flux tower site (lines 198-200) and a figure and text describing comparison of regional total basal area to the Results. These additions are also now reflected in the conceptual figure (Fig. 1). We also added treatment of the need to get composition and stocks/fluxes correct in the Discussion.

L360--361, "unrealistic benchmark" --- I think this is a bit too pessimistic. To me, an exciting future research direction in land surface modeling is to combine calibration based on functional responses and ecological patterns (which is very effectively demonstrated in this study) with data-driven initial conditions and iterative state data assimilation.

We agree that this is an exciting future direction. We mean to say that achieving the current vegetation distribution, composition, and structure without having the time series of actual climate and disturbances/management activities is unrealistic. We edited this to read:

"Current vegetation distributions are the result of particular sequences of climate, disturbances, and dispersal events across millennia (Jackson et al., 2009). Therefore, without observations of realized disturbances (including land management), and their representation in the model, a global model may not be able to precisely replicate the spatial patterns of vegetation structure and distribution from observations. Functional relationships among climate, disturbances, and vegetation distributions may provide a more meaningful benchmark"

L374--376, "driven by available light and water and the presence of fire" --- Based on the background research cited here, I am prepared to accept that this is correct in this system. However, related to my comment in the introduction, what about nutrient (e.g., N, P) limitation? Is there any representation of this in the version of CLM-FATES used here? If not, that is an important ecological question that this modeling setup is fundamentally unable to address. I might consider devoting at least a sentence or two (here or elsewhere) on the ideas about missing or incorrect model assumptions and how they determine what we can and can't learn from models (e.g., Medlyn et al. 2015, DOI: 10.1038/nclimate2621.)

We agree and have added text to the Discussion.

"Ecological expectations would then be developed for each site-PFT combination to filter the ensemble of potential PFT definitions. These expectations, and their implications, depend on the processes and ecological mechanisms represented in

the model (Medlyn et al., 2015).  If, for example, nutrient limitation has a strong influence on community composition but is not represented in the model, it would be important to assess the filtered parameterizations to understand which mechanisms are compensating to achieve the expected composition.  The filtered parameterizations can be evaluated across a larger domain with gradients of climate and soils to determine if additional parameter, or model, modifications are necessary before investing in global simulations. "

L407, "explicit representation of the flow of water" --- Two studies that may also be relevant here are Meunier et al. 2020 J. Ecol (DOI: 10.1111/1365-2745.13540) and Xu et al. 2016 (DOI: 10.1111/nph.14009).
Thank you, we have included these.

---

## Author Comment (AC2)

General comments

This manuscript describes an interesting study that attempts to efficiently parameterize the FATES vegetation demography model in novel ways. The authors start with extant trait observations which are filtered over a number of steps for parameter combinations that produce ecologically realistic forests in which the trait combinations conform to a priori expectations in relation to each other and driving data. This is highly interesting for reasons well described in the text, and I agree with Reviewer 1 about its importance for the broader field and the generally high quality of the presentation and text.

There are a few things that could be improved (see short list below). Specific spots in the text are occasionally awkward or not well integrated; a few of the figures should be tweaked or perhaps re-thought; and parts of the introduction and discussion could be tightened with little loss.

In summary, this is a really interesting approach to a hard problem that should be of wide interest to land modelers generally, and that provides a framework to build on for vegetation demographic models specifically. It needs only minor to moderate revisions before final publication.

Specific comments

    Lines 16-17: not sure this sentence ("Composition is...") is needed
We agree and removed the sentence.

    53-61: this paragraph feels disconnected from rest of the introduction
In order to better connect to the previous paragraph, the first sentence now reads "Prior research suggests that the model parameters that are most important in determining composition are likely to vary according to the model's representation of the primary constraints on plant growth (Nemani et al., 2003) and disturbance regimes."

    Figure 1: this is great—thank you. Extremely helpful in following a moderately complex analysis
Thank you.

    125: "combines observations"
Corrected

    141-142: this (specified rank correlation) is unclear; expand a bit?
We edited his section to read:
"We then ordered the sampled parameter values to maintain specified rank correlation between different parameters (Xu and Gertner, 2007;Iman and Conover,

1982).  The rank correlation matrix was calculated based on observed trait values for each PFT.  Samples for each parameter were drawn from a distribution defined by the observations, such that pairings of samples between parameters maintain the specified rank correlation (Iman and Conover, 1982)."

   Figure 2 is really cool. Hard to see light grey versus dark grey though
The 72-member ensemble points were missing from this figure.  They are now plotted and the contrast between light and dark grey is apparent.

   Figure 3 caption: "green lines"?
Changed to "colored lines"

   Figure 4: why aren't the canopy and understory outer rings grouped next to each other? I.e. blue canopy -> red canopy -> red understory -> blue understory as one goes around the outer circle
We use this arrangement because we think that showing the overall composition of pine vs cedar is the most important piece of information in this figure, given that the focus of the manuscript is on forest composition, and that canopy position is secondary.

   271: break into two sentences for readability and correct grammar
Corrected as suggested.

   315-, 334-: well described and summarized
Thank you.

   325: kind of circular...perhaps reword
Edited to:  "However, these methods do not ensure that simulated composition, even when accurate, is a result of the mechanisms that determine competitive outcomes and drive composition (Williams et al., 2009)."

   345-349: this could be expanded. How onerous *was* the computational cost? In the future would multi-site, even complete landscape, workflows be better?
FATES dramatically increases the cost.  We added text noting this, and acknowledging additional site simulations may help: "CLM-FATES simulations are approximately five to six times more computationally expensive than big-leaf CLM simulations.  Even so, selecting additional site locations stratified by environmental variables may be beneficial, particularly when developing more than two PFTs."

   359-361: a critical point but could be expanded on; is it truly unrealistic, given realistic driving data?
If the long-term driving datasets of climate and disturbance timing and magnitude are available, we can expect the models to replicate current vegetation distributions. We edited this to read:
"Current vegetation distributions are the result of particular sequences of climate, disturbances, and dispersal events across millennia (Jackson et al., 2009). Therefore,

without observations of realized disturbances (including land management), and their representation in the model, a global model may not be able to precisely replicate the spatial patterns of vegetation structure and distribution from observations. Functional relationships among climate, disturbances, and vegetation distributions may provide a more meaningful benchmark."